# PAS Domain-Containing Chemoreceptors Influence the Signal Sensing and Intestinal Colonization of *Vibrio cholerae*

**DOI:** 10.3390/genes13122224

**Published:** 2022-11-27

**Authors:** Rundong Shu, Chaoqun Yuan, Bojun Liu, Yang Song, Leqi Hou, Panpan Ren, Hui Wang, Chunhong Cui

**Affiliations:** 1College of Life Sciences, Nanjing Agricultural University, Nanjing 210095, China; 2College of Resources and Environmental Sciences, Nanjing Agricultural University, Nanjing 210095, China

**Keywords:** *Vibrio cholerae*, methyl-accepting chemotaxis protein, aerotaxis, Aer, colonization

## Abstract

Bacterial chemotaxis is the phenomenon in which bacteria migrate toward a more favorable niche in response to chemical cues in the environment. The methyl-accepting chemotaxis proteins (MCPs) are the principal sensory receptors of the bacterial chemotaxis system. Aerotaxis is a special form of chemotaxis in which oxygen serves as the signaling molecule; the process is dependent on the aerotaxis receptors (Aer) containing the Per-Arnt-Sim (PAS) domain. Over 40 MCPs are annotated on the genome of *Vibrio cholerae*; however, little is known about their functions. We investigated six MCPs containing the PAS domain in *V. cholerae* El Tor C6706, namely *aer2*, *aer3, aer4, aer5, aer6,* and *aer7*. Deletion analyses of each *aer* homolog gene indicated that these Aer receptors are involved in aerotaxis, chemotaxis, biofilm formation, and intestinal colonization. Swarming motility assay indicated that the *aer2* gene was responsible for sensing the oxygen gradient independent of the other five homologs. When bile salts and mucin were used as chemoattractants, each Aer receptor influenced the chemotaxis differently. Biofilm formation was enhanced by overexpression of the *aer6* and *aer7* genes. Moreover, deletion of the *aer2* gene resulted in better bacterial colonization of the mutant in adult mice; however, virulence gene expression was unaffected. These data suggest distinct roles for different Aer homologs in *V. cholerae* physiology.

## 1. Introduction

Bacteria utilize gene expression versatility to overcome the challenges of host defenses and natural stresses. Bacterial chemotaxis, the ability to migrate in response to gradients of environmental chemicals [1], is one of the essential stress response mechanisms for bacterial survival and colonization [2]. Numerous receptors are involved in regulating chemotaxis and other cellular functions [3]. Chemotaxis regulation has been well studied in *Escherichia coli*. In brief, after methyl-accepting chemotaxis proteins sense intracellular and environmental cues, the sensory information is transmitted to the chemoreceptor histidine kinase CheA and the adaptor protein CheW, and then to the flagellar motor by phosphorylating the cognate response regulator, CheY [4,5].

Chemotaxis that occurs in the presence of an oxygen gradient is known as aerotaxis. Two different aerotaxis mechanisms have been reported. The first mechanism is based on the binding of heme-containing receptors (e.g., HemAT) for extracellular oxygen detection [6,7,8,9]; the second one is based on the intracellular energy level to determine whether more oxygen is required [9]. The oxygen sensing in *E. coli* depends on the aerotaxis receptor (Aer), which has an energy-sensing module Per-Arnt-Sim (PAS) domain at its N-terminus and functions with a cofactor. The PAS domain monitors cellular oxygen levels and redox states through flavin-adenine dinucleotide (FAD) and subsequently triggers aerotactic responses [10,11].

*E. coli* possesses a single chemotaxis system. In contrast, the majority of chemotactic bacteria have multiple chemotaxis systems. *Vibrio cholerae*, the causative agent of cholera, uses complex signal transduction networks to regulate a set of chemotaxis genes during both the free-living phase in the aquatic environment and short-term infection phase in the human gut [2,12,13]. Genomic analysis revealed three clusters of core chemotaxis-related genes (clusters I, II, and III) in *V. cholerae*, but only cluster II is critical for chemotaxis under laboratory conditions [14,15]. According to cytoplasmic signal domain variations, the methyl-accepting chemotaxis proteins (MCPs) of *V. cholerae* can be classified into 44H, 40H, 36H, and 24H. Boin et al. observed that the deletion of a 40H Mlp32 (MCP-like protein 32) rendered the aerotaxis response in classic biotype *V. cholerae* [16]. Greer-Phillips et al. reported that the 36H Mlp45 hijacks the chemotaxis system and mediates the oxygen response in *E. coli*, whereas no related physiological responses were detected in *V. cholerae* [17]. 

In this study, 45 potential MCP proteins were identified in the genome sequence of *V. cholerae* El Tor C6706 [18]. We investigated six MCPs with the PAS domain and revealed their physiological functions throughout the bacterial life cycle.

## 2. Materials and Methods

### 2.1. Strains, Plasmids, and Culture Conditions

All *V. cholerae* strains used in this study were derived from O1 El Tor strain C6706 [19]. In-frame deletions of mutant were constructed by cloning the fragments flanking target genes into the suicide vector pWM91, which harbored a *sacB* counterselectable marker [20]. After double-crossover recombination, mutants were isolated from sucrose plates and confirmed using PCR. Transcriptional fusion reporters were constructed by cloning the promoter sequences of target genes into a pBBR-lux vector which contains a promoterless *luxCDABE* reporter [21]. The overexpressing plasmids were obtained by cloning the complete target gene sequence into the pSRKGm vector [22]. Strains were grown in Luria–Bertani (LB) broth aerobically at 37 °C unless otherwise stated. Antibiotics were added at the following concentrations: ampicillin 100 μg/mL, streptomycin 100 μg/mL, gentamicin 20 μg/mL, chloramphenicol 10 μg/mL (for *E. coli*) or 2 μg/mL (for *V. cholerae*). Isopropyl-β-d-1-thiogalactopyranoside was supplemented at a final concentration of 100 μg/mL.

### 2.2. Bioinformatic Analysis

Putative methyl-accepting chemotaxis proteins in *V. cholerae* C6706 genome were identified using MiST 3.0 database (https://mistdb.com/ (accessed on 6 Jul. 2019) by searching for the keyword “MCP”. Protein domains were predicted with the SMART database (http://smart.embl-heidelberg.de/ (accessed on 10 Jul. 2019). Amino acid sequences of selected proteins were aligned using MUSCLE (https://www.ebi.ac.uk/Tools/msa/muscle/ (accessed on 20 Aug. 2019). 

### 2.3. Growth Curves

Overnight cultures were washed with phosphate-buffered saline, diluted 1:100 using fresh LB medium or succinate minimal medium, and incubated at 37 °C with shaking at 180 rpm. OD_600_ was measured at the indicated time points. Three independent experiments were performed.

### 2.4. Swarming and Swimming Motility Assays

A swarming assay was performed on soft-agar (0.3% agar) M9 minimal medium supplemented with succinate (50 mM) as a carbon source [23]. Mucin (200 mg/mL) and bile salts (0.5%) or KCl (200 mM) were used as chemoattractants. A sterile toothpick was touched lightly on a bacterial colony and poked into the bottom of a soft-agar plate. Diameter was measured after 24 h incubation at 30 °C. To evaluate swarming motility under anaerobic conditions, 50 mM KNO_3_ was added as an alternative electron acceptor. Plates were incubated in an anaerobic chamber at 30 °C for 4 d. Three independent experiments were performed.

### 2.5. Air Trap Assay

A modified air trap assay was performed based on the method described [24]. Bacterial cultures were grown overnight at 30 °C on the LB plates and suspended in phosphate-buffered saline at an OD_600_ of 1.0. Approximately 100 μL of culture was injected by syringe into an LB soft-agar tube (0.1% agar). The tube containing a Pasteur pipette that protruded 2 to 3 cm from the LB medium was sealed with a sterilized cap. The bacterial sample was deposited into the LB medium but outside of the Pasteur pipette. After inoculation, 2 mL paraffin oil was added to the LB medium (carefully avoiding falling into the Pasteur pipette). After 24 h incubation at 30 °C, 100 µL of the air-medium contact surface in the Pasteur pipette was recovered, and the CFU were counted on the LB plates. Three independent experiments were carried out.

### 2.6. In Vitro Assays for Virulence Genes Expression 

Overnight culture of bacterial strains containing a promoterless *luxCDABE* transcriptional fusion plasmids of virulence gene *tcpA* or *tcpP* were transferred into the virulence-inducing AKI medium (1:10,000) [25] and incubated at 37 °C until OD_600_ reached 0.2 at various oxygen levels, including aerobically (shaking), micro-aerobically (standing), and anaerobically (chamber, standing). Luminescence was measured and normalized against OD_600_. Three independent experiments were performed.

### 2.7. Biofilm Assay

Bacterial strains were grown overnight and inoculated 1:100 in 10 × 75 mm borosilicate glass tubes containing 800 µL fresh LB broth and incubated at 37 °C without shaking. After 24 h incubation, the contents of each tube were gently removed. Tubes were stained using 0.1% crystal violet for 5 min. Biofilm was dissolved in dimethyl sulfoxide and quantified by measuring the OD_570_ [26]. Three independent experiments were performed.

### 2.8. Adult Mouse Colonization Assays

The colonization assays using an adult mouse model were performed as previously described [27]. Briefly, overnight cultures of wild-type (*lacZ*^−^) and mutant (*lacZ^+^*) strains were mixed equally and approximately 10^8^ *V. cholerae* cells were administered by intragastrical gavage to 6-week-old female CD-1 mice that were pretreated with streptomycin. Fecal pellets were collected at the indicated time points, resuspended in phosphate-buffered saline, serially diluted, and spread on LB plates containing X-Gal. The competitive index was calculated as the recovery ratio of mutant colonies to wild-type colonies divided by the input ratio of mutant to wild type. Five mice were used in each experiment.

## 3. Results

### 3.1. Analysis of Putative MCP Sequences

MCPs are one of the first components in chemotaxis to sense and adapt to excitation. Bioinformatic analysis revealed 45 putative MCPs in *V. cholerae* C6707 genome, and six of them possess one or two PAS domains at the N-terminus (VCA0658, VCA0988, VCA1092, VC0098, VC1406, and VCA0864) (Figure 1A). PAS domains regulate the function of many intracellular pathways in response to oxygen tension, redox potential, or light intensity [16,28], and proteins with PAS domain may play important roles in sensing both extrinsic and intrinsic stimuli. Therefore, we selected these six genes for further experiments. Sequence comparison between several Gram-negative bacteria shows that these PAS-domain-containing genes are homologs of the *aer* gene [29] (Figure 1B). We named *aer2*, *aer3*, *aer4*, *aer5*, *aer6*, and *aer7*.

### 3.2. Aer2 Mutant Is Impaired in Aerotaxis

To investigate the role of these putative *aer* genes in *V. cholerae* motility, we first constructed an in-frame deletion mutant of each *aer* gene. The growth of these *aer* mutants were comparable to that of wild-type strain in LB medium (Appendix A) and M9 minimal medium (Appendix A). Moreover, these mutants demonstrated similar migration ability in the swimming motility assay (Appendix A). These results suggest that none of these genes are crucial for cell viability.

The swarming ability of these mutants was further determined using the swarming motility assay. The diameter of the swarm ring was measured after 24 h incubation. Only the ∆*aer2* mutant displayed a significant defect when compared to the wild type (Figure 2A,B), indicating that the motility disadvantage of the ∆*aer2* mutant is limited in the swarming behavior. We then constructed a complementary strain by introducing the recombinant plasmid p*SRK-aer2* into the ∆*aer2* mutant strain (hereafter named ∆*aer2*^C^). Figure 2C showed that the overexpression of *aer2* not only restores the swarming motility, but even enhances aerotaxis compared with the phenotype of wild type carrying the empty vector plasmid (Figure 2C). These findings are consistent with those in *V. cholerae* classic biotype O395 and *E. coli* K12 [23,30], indicating that *aer2* is essential for bacterial swarming motility.

In anaerobic respiration, aerotaxis receptors may employ other oxide compounds, such as sulfate, nitrate, sulfur, and fumarate, as electron receptors instead of oxygen [23]. To assess swarming behavior under anaerobic conditions, KNO_3_ was added to the medium as an alternative electron acceptor [11]. Under anaerobic conditions, the results of swarming motility assays were similar in the ∆*aer2* mutants and wild type (Figure 2D), implying that the difference in the outward movement of these two strains on semi-solid succinate plate in aerobic conditions was in response to oxygen gradient. 

Although little difference was detected in the swarming motility of the other five *aer* mutants (*aer3, aer4, aer5, aer6,* and *aer7*), we attempted to elucidate whether these *aer* homologs may cooperate with *aer2* and contribute to aerotaxis. For this purpose, we constructed a deletion mutant ∆*aer234567* and compared the swarming motility between ∆*aer2*, ∆*aer234567*, and wild type (Figure 3A). The swarm ring diameter of ∆*aer234567* was approximately 0.837 cm, which was comparable to that of ∆*aer2* (0.887 cm). The swarm ring diameter of both the mutants was markedly reduced compared with that of the wild type (1.1 cm). These results suggested that these five *aer* genes (*aer3*, *aer4*, *aer5*, *aer6*, and *aer7*) were not involved in the oxygen sensing process. An air trap assay was performed to further validate the results by determining the bacterial count on the medium-air contact surface of the Pasteur pipette [30,31]. Compared to the wild-type, the absence of *aer2* resulted in a 58-fold decrease in CFU of the ∆*aer2* mutant (Figure 3B). The additional knockout of the other five *aer* homologs did not decrease the bacterial count further. Taken together, these results suggest that Aer2 is the only aerotaxis receptor of *V. cholerae* C6706 among the MCPs with the PAS domain under laboratory conditions.

### 3.3. Each Aer Gene Is Involved in Different Chemotactic Responses

Our earlier experiments revealed that *aer2* was a specific aerotaxis receptor when oxygen functions as an attractant (Section 3.2). None of the other five putative *aer* genes exhibited oxygen-sensing-related phenotype; therefore, we speculated that they may be involved in the response to other chemical agents. *V. cholerae* is an enteric pathogen with complex pathways activated in response to gut signals. Here, we selected mucin, bile salts, and potassium chloride as chemoattractants. When mucin was used as a chemoattractant, compared to the chemotactic ring diameter of the wild type, ring diameters of the *∆aer2* and *∆aer234567* mutants were significantly decreased (Figure 4A), In contrast, no difference was observed between the ring diameters of the other *aer* mutants and that of the wild type. However, the phenotypes were different when bile salts served as a chemoattractant (Figure 4B); the migrated diameters of ∆*aer2*, ∆*aer3*, ∆*aer6*, ∆*aer7*, and *∆aer234567* mutants significantly increased. The chemotactic rings of ∆*aer6* and ∆*aer234567* were approximately two-fold larger than that of the wild type. Chemotaxis of the ∆*aer4* mutant was inhibited notably. In contrast, bacteria were less chemotactic to potassium chloride, and the chemotactic ring diameter of the *aer* mutants were similar to that of the wild type (Figure 4C). The results revealed that Aer transducer receptors are required for chemotaxis toward mucin and bile salts, and different *aer* homologs contribute to different chemotactic responses.

### 3.4. Aer Contributes to the Pathogenesis of V. cholerae C6706

Complex chemosensory systems control multiple functions in bacteria, such as swarming [32], biofilm formation [32], autoaggregation [33], and interactions with the host [34]. To test whether these *aer* genes are important for the biological functions of the C6706 strain, we compared the phenotype of wild-type and *aer* mutants in different physiological contexts. Biofilms are multicellular bacterial communities that adhere to a surface or interface, and biofilm-mediated attachment is important for *V. cholerae* survival in the environment [2,35,36]. We examined whether the deletion of any *aer* gene affects the C6706 biofilm formation. The result showed that the absence of these *aer* genes did not significantly affect biofilm formation (Figure 5A, blue). Notably, overexpression of the *aer6* and *aer7* genes enhanced biofilm formation (Figure 5A, orange), indicating that these genes may contribute to the biofilm developmental process in some unknown specific conditions.

Next, we investigated whether these *aer* genes play any role in the pathogenesis of C6706. Although the earlier results suggested that *aer2* plays a critical role in the aerotaxis of C6706 independent of the other five *aer* genes, we cannot exclude that these *aer* genes may cross-talk when the virulence genes are activated. First, we detected the influence of the *aer* genes on the expression of *tcpP* and *tcpA* [37], the main virulence factors of *V. cholerae*, in the induced AKI medium. Wild-type, ∆*aer2*, and ∆*aer234567* mutants harboring various virulence gene reporter plasmids [38] were grown under the aerobic (shaking), microaerobic (standing), and anaerobic (chamber, standing) conditions (Figure 5B). Consistent with a previous report [38], the expression of the virulence genes was dependent on the oxygen concentration, while strains with mutations in the *aer* homologs demonstrated similar expression levels as wild type under similar conditions. These observations suggested that these *aer* genes have a negligible effect on virulence gene expression under stable oxygen concentrations.

We further utilized an adult mouse competition model to test for the colonization of *∆aer2* and *∆aer234567* mutants. Equal volumes of wild-type and ∆*aer2* or *∆aer234567* mutant were mixed and intragastrically administered to mice. Fecal samples were collected at the indicated time points to determine the output ratio of the mutant to the wild type. As shown in Figure 5C, the wild type was outcompeted by the *∆aer2* mutant. At day 4 post-inoculation, the competitive index was up to 10. However, colonization difference was not observed between the wild type and ∆*aer234567* in this mouse model. These data suggest that *aer2* also plays an important role *in vivo*, which may not be associated with virulence gene expression. Other Aer receptors function differently to adapt to the complex intestinal environment.

## 4. Discussion

Bacterial directed movement in response to natural gradients of chemical stimuli is defined as chemotaxis; one of the special forms is called aerotaxis when oxygen serves as an attractant or repellent. Both chemotaxis and aerotaxis contribute to bacterial growth, survival, and pathogenesis [4,11,39,40]. Aer consists of the cytoplasmic PAS domain that binds an FAD cofactor, and acts as a redox sensor to track changes in the concentration of oxygen and other electron acceptors in the environment [41]. Pokkuluri et al. reported that *Geobacter sulfurreducens* uses heme-containing PAS sensors to send out signals in anaerobic environments [42]. Watts et al. demonstrated that Aer-2 chemoreceptor of *Pseudomonas aeruginosa* also has a PAS domain to mediate repellent responses to oxygen, carbon monoxide, and nitric oxide [43]. In this study, six MCPs (VCA0658, VCA0988, VCA1092, VC0098, VC1406, and VCA0864) were selected from 45 MCPs of *V. cholerae* El Tor biotype C6706. All these six MCPs possess one or two PAS domains at N-termini, and they were identified as homologs of Aer protein in *E. coli* and *P. aeruginosa (*Figure 1A). Growth and swimming defects were not detected in any of *aer* homolog mutants, suggesting that Aer are not important for *V. cholerae* in the regular culture conditions. However, in the swarming motility assay and air trap experiment, the ∆*aer2* mutant displayed a significant defect in the aerotaxis, and there were no functional interactions between these six putative Aer receptors (Figure 2 and Figure 3). This implied that *aer2* is directly involved in the aerotaxis of *V. cholerae* El Tor O1 biotype C6706. Boin et al. reported that Aer-2 (VCA0658) but not Aer-3 (VCA0988) actively mediates an aerotaxis response in the classical biotype of *V. cholerae* O395N1 [23]; our results are consistent with this observation. 

Mucin, bile salts, and potassium chloride are representative compounds that affect energy taxis in bacteria. Although Na^+^ and K^+^ are the major drivers of flagellar motility and crucial environmental factors for *V. cholerae* survival [44], they do not act as the signals for Aer-mediated chemotaxis (Figure 4C). It has been reported that both mucin and bile are chemoattractants for *V. cholerae* [45,46]. Esmeralda et al. demonstrated that accessory colonization factor *acfC* (VC0841) mediates the chemotaxis to mucin [47]. Nishiyama et al. found that Mlp37 (VCA0923) was an important chemoreceptor for bile salts [46]. In this study, we also demonstrated that Aer receptors were involved in the chemotaxis with mucin and bile salts as chemoattractants, but each *aer* homolog contributes to different chemotactic responses (Figure 4A,B). Mucin is considered as barrier for the colonization of bacteria in the intestinal epithelium [48,49]. Owing to their detergent-like properties, bile salts present in the digestive fluid can destabilize bacterial membranes, thereby altering cellular equilibrium [50]. Our results confirmed that the Aer receptor likely facilitates the ability of *V. cholerae* to find a suitable ecological niche in the host intestine and its subsequent survival.

After responding to different stimuli by chemotaxis and aerotaxis, bacteria regulate the target genes accordingly and change their collective behaviors. For example, PAS domain-containing proteins play an important role in the biofilm development of *P. aeruginosa* [40]. Biofilm is critical for bacterial pathogenesis in colonization in their environmental habitats as well as host gut [51,52]. Our results showed that Aer6 and Aer7 were able to enhance the biofilm formation capability of the C6706 strain (Figure 5A).

The virulence gene expression and colonization of *V. cholerae* in the host are closely linked to aerotaxis [53,54], but the possible role of chemotaxis in establishing productive infections is still under investigation. Millet et al. demonstrated that strains with mutantation in chemotaxis genes, such as *vspR*, *pomA*, or *cheA*, hypercolonized the intestine of infant rabbits [55]. Kamp et al. found that most chemotactic genes are dispensable for *V. cholerae* infection, but play a crucial role in survival in pond water [52]. Our *in vitro* results showed that *aer* genes are not involved in virulence gene expression under stable oxygen concentrations. However, ∆*aer2* had a colonization advantage whereas *∆aer234567* colonized similarly to the wild type in the small intestine of adult mice (Figure 5C). These four *aer* homologs (*aer3*, *aer4, aer6*, and *aer7*) play different roles in various biological process. We speculate that deletion of the key gene involved in aerotaxis, *aer2*, enhances colonization. Although each *aer* receptor may play a certain physiological role to balance its persistence in the host, further research is required to characterize the mechanism. To conclude, our results revealed that PAS domain-containing chemoreceptors are important for *V. cholerae* physiology.

## Figures and Tables

**Figure 1 genes-13-02224-f001:**
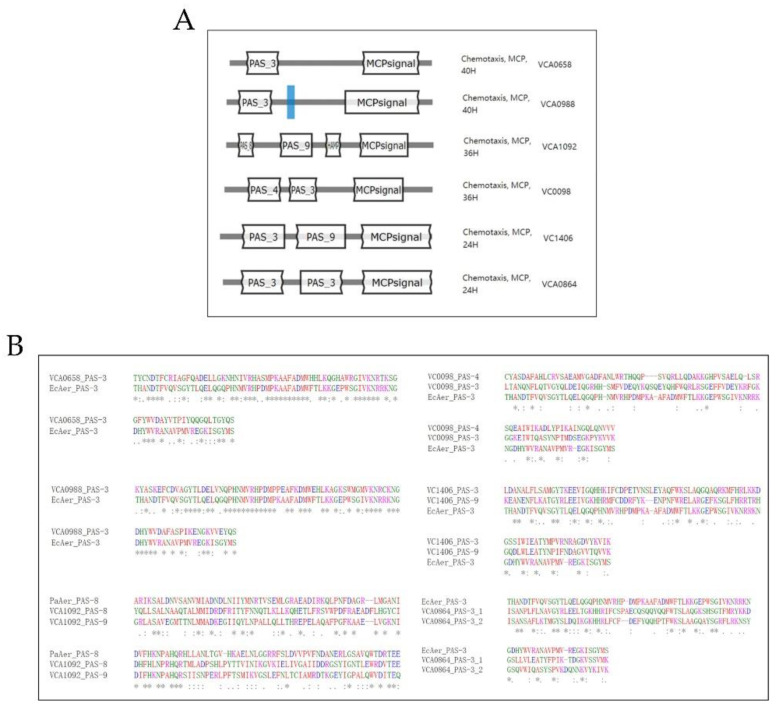
In silico analysis of the six MCPs with PAS domain in *V. cholerae* C6706 genome. (**A**) Predicted domain structure and type using MiTS 3.0. (**B**) Amino acid sequence alignment of *aer* homologs using MUSCLE. Residue conservation between groups are labeled with asterisk (*), colon (:), and dot(.) from high to low similarity.

**Figure 2 genes-13-02224-f002:**
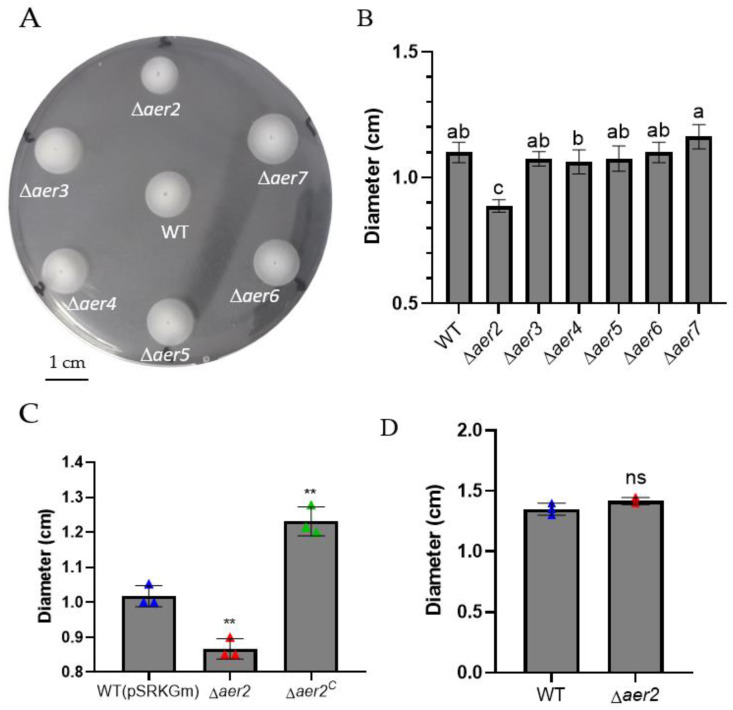
Swarming motility assay. A swarming motility assay was performed using *V. cholerae* C6706 wild-type (WT), *aer* mutants, and the *aer2* complementary strain (∆*aer2*^c^) on succinate soft agar plate (**A**), and the average swarm circle diameters were measured (**B**). Different letters in a single column indicate a significant difference between treatments (One-way ANOVA; *p* < 0.05). (**C**) Swarm ring diameter of wild-type, ∆*aer2* mutant, and *aer2* complementary strain. Plates were incubated at 30 °C for 24 h. ∆*aer2* mutant exhibited limited swarming motility, and ∆*aer2^c^* had swarming motility superior to that of the wild type. (**D**) Swarming motility assay in anaerobic conditions. KNO_3_ was added to the medium as the alternative electron acceptor. Plates were incubated in an anaerobic chamber at 30 °C for 4 d before the swarm ring diameter was measured. **, *p <* 0.01; ns, no significance (Student *t*-test).

**Figure 3 genes-13-02224-f003:**
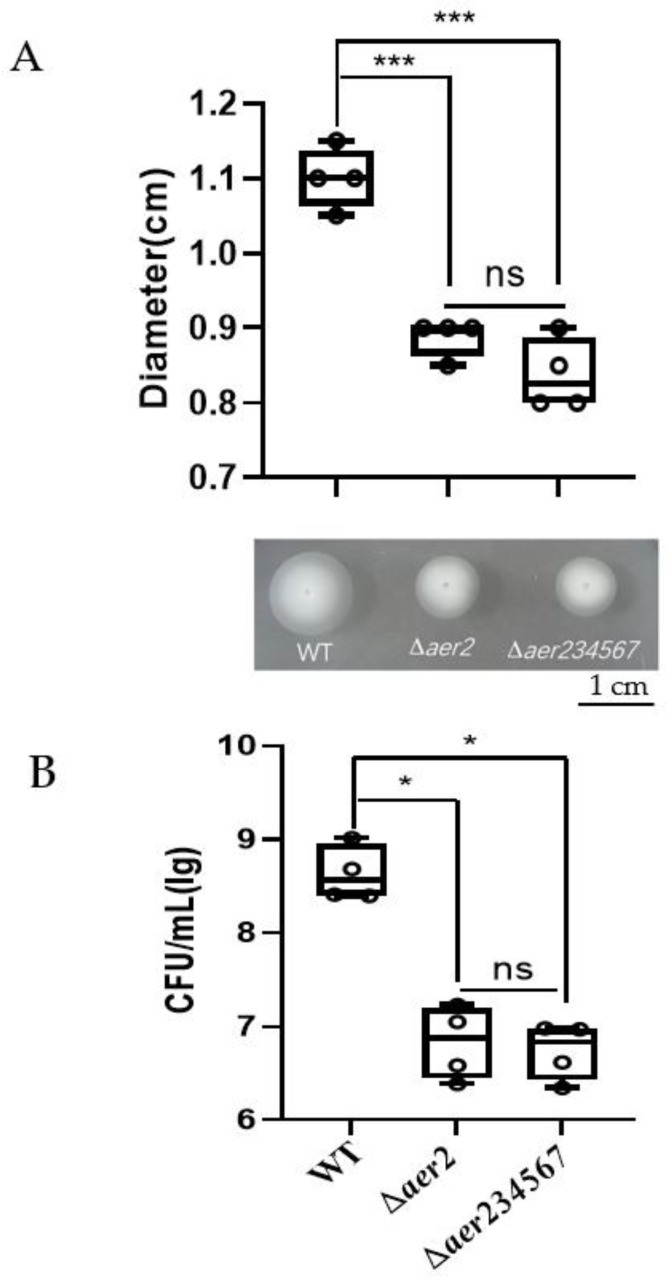
Swarming motility assay. The migration ability of WT, Δ*aer2*, and Δ*aer234567* mutants using a soft-agar plate after 24 h incubation at 30 °C (**A**), or by calculating CFU on the medium-air contact surface of the Pasteur pipette in the air trap assay (**B**). *, *p* < 0.05; ***, *p* < 0.001; ns, no significance (Student *t*-test).

**Figure 4 genes-13-02224-f004:**
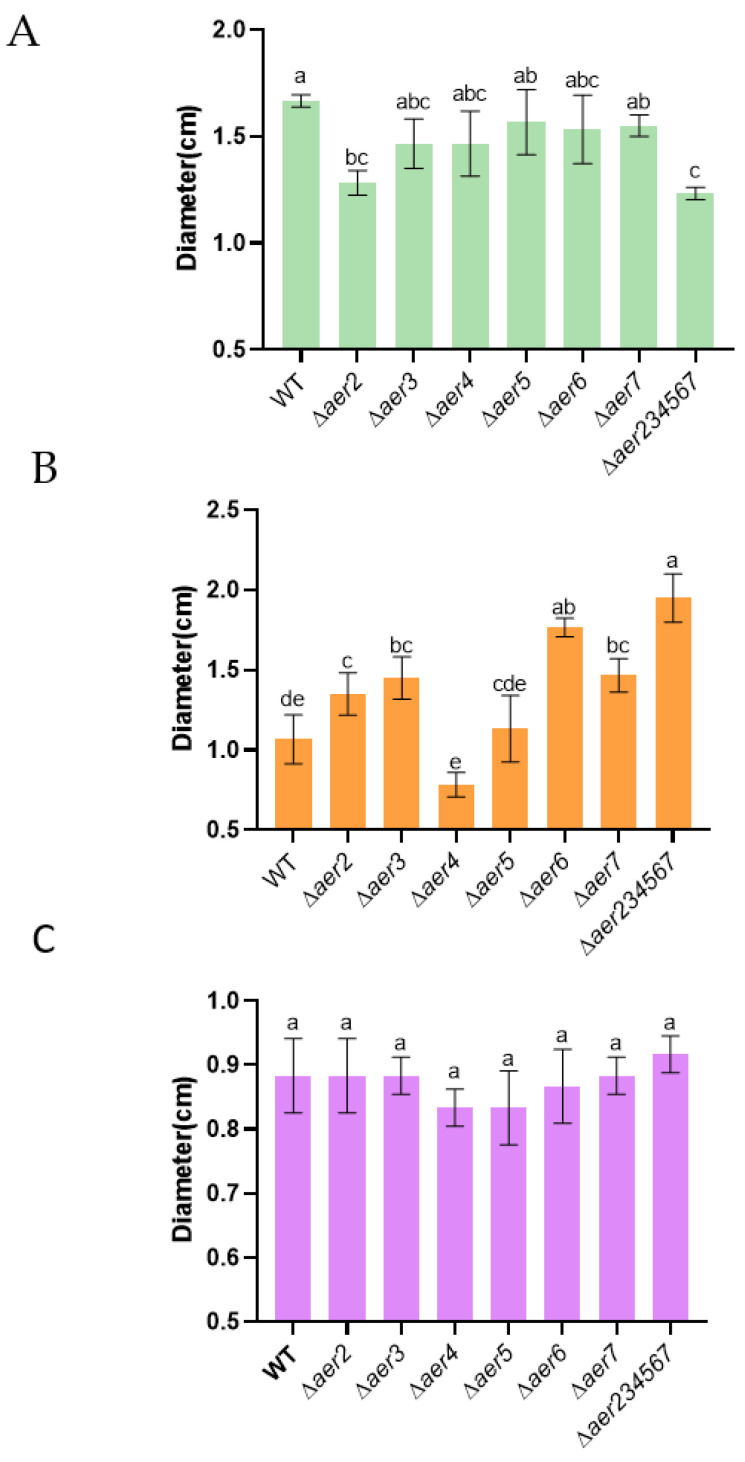
Chemotaxis assay of the mutant and wild-type strains. Chemotaxis behavior was observed on soft-agar plates with 200 mg/mL mucin (**A**), 0.5% bile salts (**B**), and 200 mM KCl (**C**) as attractants. Bacterial cultures were incubated for 24 h at 30 °C. Different letters in a single column indicate significant differences between treatments (One-way ANOVA; *p* < 0.05).

**Figure 5 genes-13-02224-f005:**
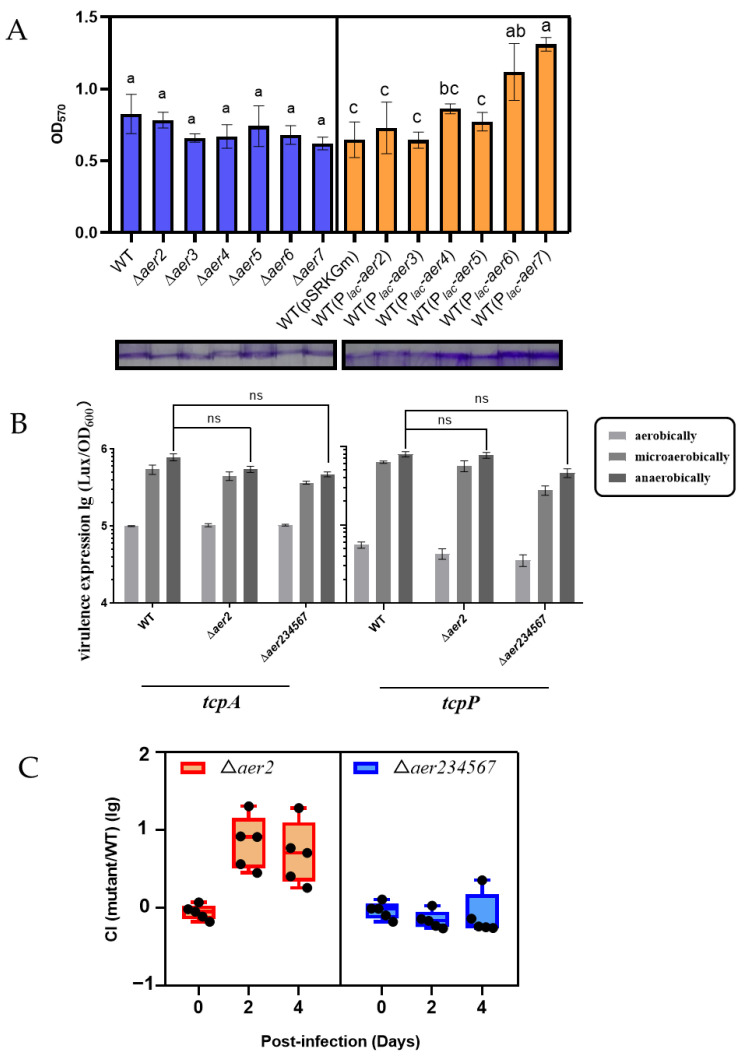
Effect of the *aer* genes on the colonization and pathogenesis of *V. cholerae* C6706. (**A**) Biofilm of mutant strains (blue bar) and complemented strain (orange bar). Overnight bacterial cultures were inoculated in the LB medium and grown at 37 °C for 24 h. Tubes were stained with crystal violet. Different letters in each column indicate significant differences between treatments (One-way ANOVA; *p* < 0.05). (**B**) Virulence gene expression under different oxygen conditions. Overnight bacterial cultures were inoculated into the virulence-inducing AKI medium (1:10,000). Cultures were grown aerobically (shaking), micro-aerobically (standing), and anaerobically (chamber, standing) at 37 °C. Luminescence was measured and normalized to OD_600_. ns, no significant difference (Student *t*-test). (**C**) Bacterial colonization in adult mice. Mice were intragastrically administered with 10^8^ cells of a 1:1 mixture of wild type and Δ*aer2* or Δ*aer234567* mutant. CFU of fecal pellets were determined using a selective medium. The competitive index was calculated as the ratio of mutant to wild-type colonies normalized to the input ratio. Horizontal line: mean of values obtained for five mice.

## Data Availability

Not applicable.

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
