# Peer review of "PAS Domain-Containing Chemoreceptors Influence the Signal Sensing and Intestinal Colonization of Vibrio cholerae"

_genes, 2022, doi:10.3390/genes13122224_

Round 1

Reviewer 1 Report

It was cited that a group of MCP-like proteins were studied to determine their functions in the life cycle of Vibrio cholera, and how these functions relate to Aerotaxis.  You found a number of aer genes that were selected as possible candidates receptors for sensing oxygen.

Your data suggests only one gene, aer 2, may have Aerotaxis function.  All of the other genes do not seem to.  Now you discussed a number of other chemotaxis activities with the other aer genes, which appeared to be the bulk of your paper.  Yet the main objective of your paper was supposed to be Aerotaxis analysis.  If the main objective of this Vibrio cholera study was on various chemo/aerotaxis responses to start, the flow of your data and results would make more sense.

Reviewer 2 Report

General comments:

The authors present some functional data on 6 different methyl-accepting chemotaxis proteins (MCPs) encoded by V. cholerae C6706, with a focus on one of the proteins Aer2 which is described as having a function in aerotaxis. Results indicating a role of two further MCPs, Aer 6 and 7, in biofilm formation and chemotaxis to mucin and bile salts, are also shown. Although the functions are not described or studied in much detail, the study adds value in that it lays the foundation for future studies by these and other researchers. The text needs substantial English language revision. For now, the reader is forced to guess the meaning of sentences and statements throughout the whole manuscript.

Specific comments:

Line 16: PAS: please write all abbreviated words in full when used first time Line 56: change “genomes” to “chromosomes”

Line 158: the link is to the shopping cart. What exactly was used/bought?

Line 176 (figure 1 legend): change MITS3 to MiST 3.0 Figure 2B, 4 and 5: The different letters at the top of columns which are supposed to indicate significant differences (p < 0.05) are very confusing, it is not clear what is compared to what. Please find a way to show significant differences in a clearer way.

Reference list is not consistent, often only initials of authors are shown.

Supporting figure S1 is missing.
